# A Comparison of Three Approaches for Larval Instar Separation in Insects—A Case Study of *Dendrolimus pini*

**DOI:** 10.3390/insects10110384

**Published:** 2019-11-02

**Authors:** Lidia Sukovata

**Affiliations:** Department of Forest Protection, Forest Research Institute, 3, Braci Leśnej St., Sękocin Stary, 05-090 Raszyn, Poland; L.Soukovata@ibles.waw.pl; Tel.: +48-22-715-3832

**Keywords:** Brooks-Dyar’s rule, clustering, Crosby’s growth ratio, *Dendrolimus pini*, frequency distribution, head capsule width, instar, kernel density

## Abstract

The number of larval instars is important from both theoretical and practical perspectives. Three previous studies based on head capsule widths (HCWs) have suggested that *Dendrolimus pini* larvae pass through seven instars, but the estimated HCW means differed greatly. Various methods are available for determining the number of instars; however, these methods have not been compared on the same dataset. Therefore, the main goal of this study was to compare three approaches for instar separation in *D. pini* larvae: visual approach followed by non-linear least squares (NLLS) estimation, kernel density estimation (KDE) followed by NLLS, and model-based clustering. Two criteria were used to assess whether the resulting instar separations adhered to Brooks-Dyar’s rule: Crosby’s growth rule and a coefficient of determination indicating the goodness of fit of a straight line to the ln-transformed mean HCW of the respective instars. Our results showing that *D. pini* larvae pass through eight instars differed greatly from reports in the literature. The best results were obtained by KDE followed by NLLS. For proper instar separation, both criteria of Brooks-Dyar’s rule must be met.

## 1. Introduction

The pine-tree lappet moth, *Dendrolimus pini* L. (Lepidoptera: Lasiocampidae) is one of the most important defoliating insect species of Scots pine *Pinus sylvestris* L. in Europe and Asia [1]. In Poland, outbreaks of this species have become more frequent and widespread in the last two decades than in the previous 60 years [2], and might have been caused by climate change. The most recent outbreak in 2012–2014 covered the largest area since the 1950s, reaching 131,000 ha in 2013. Climate change seems to be responsible for an increase in the *D. pini* populations in Scotland [3] and an isolated outbreak in the Stockholm archipelago [4]. One of the possible effects of climate change on insects is increased voltinism [5,6]. A rapid shift from univoltinism to bivoltinism due to climate change was shown in *Dendrolimus spectabilis* (Butler) in Vietnam [7]. Depending on the country and region, another pine moth species, *Dendrolimus punctatus* (Walker) may have up to five generations per year [8,9]. *D. pini* has one generation per year in warmer climatic conditions, e.g., in Poland the flight period is from July to August, the larvae feed on needles until the first early frosts in late autumn, then descend to the forest litter for overwintering; from the early spring of the following year, they continue feeding until they pupate in June [10,11]. However, in cooler regions, the development of one generation takes two years, with the larvae overwintering twice, e.g., semivoltine *D. pini* in Scotland [12]. The probability of outbreaks increases with increasing voltinism, reproductive synchrony, synchrony with host plants, etc. Therefore, it seems that instar ratios in the larval population could serve as an indicator of population status.

*D. pini* larvae are thought to pass through six (males) or seven (females) instars [11,13,14]. However, the mean widths of the larval head capsules for each instar vary substantially across different studies, e.g., from 0.7 mm to 1.27 mm in the first instar, from 3.33 mm to 5 mm in the sixth instar, and from 5.01 to 6 mm in the seventh instar (Table 1). In addition to the existence of disparities in the mean head capsule width, no width intervals were given for any instar, which makes instar determination rather difficult.

Currently, different methods are available for determining the number of larval instars. The most popular method is the analysis of the frequency distribution based on a visual examination of the histograms; other methods include kernel density estimation (KDE) and cluster analysis based on k-means or Gaussian mixture models [15,16,17,18]. The results are usually evaluated for adherence to Brooks-Dyar’s rule and Crosby’s growth rule [19]. The former rule states that there is a geometric increase in the size of a sclerotized structure between consecutive instars; if any deviations from a straight line are observed when the data are plotted on a logarithmic scale, it is assumed that an instar is missing [19]. Crosby’s growth rule clarifies that a deviation is significant when there is a 10% or greater difference between two consecutive Brook-Dyar’s ratios [19] (see more details in Materials and Methods). Although Brooks-Dyar’s and Crosby’s rules are used by most researchers, it is uncertain whether the methods used thus far for determining the number of instars and the means and limits of head capsule widths for each instar give comparable results.

Therefore, the goals of this study are (1) to compare three methods for instar determination by conducting histogram analysis followed by non-linear least squares (NLLS) parameterization, KDE followed by NLLS, and model-based clustering, (2) to determine the number of instars in *D. pini* larvae, (3) to clarify the mean width of the head capsule and determine the width ranges for different instars, and 4) to evaluate whether the mean head capsule widths obtained from different methods in our study and those presented in the literature adhere to Brooks-Dyar’s rule.

## 2. Materials and Methods

### 2.1. Head Capsule Measurements

The head capsules of *D. pini* larvae were measured in 2010–2015. Three types of larvae were used to ensure that all instars were represented: (1) larvae hatched from eggs in the laboratory, (2) larvae hibernating in forest litter in different regions of Poland and collected during autumnal searches by foresters for population density assessment, and (3) larvae collected from the crowns of trees cut onto canvas sheets in late spring for additional population density assessment. In addition, the collection of larvae from various conditions (including laboratory rearing) enabled the coverage of a wide range of variability in head capsule widths, so that the probability of encountering a non-matching population in nature would be relatively low.

All larvae from the first group and different portions of larvae from the two latter groups were reared in the laboratory, in different years, under the room conditions, in groups of up to 50 larvae at younger, i.e., first through fourth, instars, and up to 30 larvae at older instars on Scots pine twigs maintained in 45 × 30 × 23 cm or 60 × 30 × 21 cm glass aquaria with the two smaller opposite sides covered with a net to ensure air flow. The shed head capsules were collected during each food change and cleaning procedure and then measured. After pupation of the first individuals, the head capsules were measured directly on the larvae because the head capsules were often broken after molting from the last instar to a pupa, and their accurate measurement was rather difficult. During the pupation period, the larvae were checked every one to two days, which enabled easy identification of the freshly molted individuals and measurement of their new head capsules.

In total, there were 1107 records of the maximum widths of the head capsules for the larvae hatched from the eggs and reared in the laboratory (hereafter called the ‘lab’ group), 3548 records for the larvae collected in the field and reared in the laboratory (‘field_lab’ group), and 3534 records for the larvae collected in the field, in which the head capsules were measured once (‘field’ group). The measurements were taken with a Zeiss SteREO Discovery.V8 stereoscopic microscope (Carl Zeiss sp. z o.o., Poznań, Poland) with a micrometer and were converted into millimeters (to the nearest 0.01 mm), while taking into account the total level of microscope magnification at which the measurements were made.

### 2.2. Data Analysis

First, the histograms for the head capsule widths from three groups of larvae were constructed to check if there were any substantial discrepancies in distribution patterns. The patterns were rather similar (see Results); therefore, the data from different groups were combined and, under the assumption that the head capsule width distribution of *D. pini* larvae was normally distributed, the data were analysed using three approaches.

Approach 1 (hereafter called the visual approach) was based on creating histograms for head capsule widths and visual instar separation at the lowest point between overlapping peaks. Then, the following procedure included NLLS parametrization of the estimates for the mean and variance of each dataset representing particular instars; this procedure was described in detail by McClellan and Logan [15] and Logan et al. [20]. These estimates were used to determine the lower and upper limits of the head capsule widths for each instar. A histogram was constructed with a class width of 0.12 mm, and curves fitting a normal distribution for each instar were drawn in Statistica 10 (StatSoft, Inc. 2011, Tulsa, OK, USA). NLLS analysis was conducted with the ‘nls’ function in the ‘stats’ package in the R environment (R Core Team, 2018, Vienna, Austria). All other necessary calculations were performed in Microsoft Excel 2010, v.14.0.7239.5000 (Microsoft Corporation, 2010, Redmond, WA, USA).

Approach 2 applied KDE, which is a nonparametric density estimator requiring no assumption that the underlying density function is from a parametric family [21]. Three methods were used to determine an optimal band width: (1) Silverman’s rule of thumb (hereafter SRT) [22], (2) the method of Sheather and Jones (hereafter SJ) [23], and (3) visual selection as a possible approach based on user knowledge [24]. The first two methods were tested using the ‘density’ function in the ‘stats’ package in the R environment. In the third method, different band widths ranging from 0.05 to 0.12 mm were tested by applying the Kernel Density Add-in for Microsoft Excel (Royal Society of Chemistry, Analytical Methods Committee, 2002, Cambridge, UK) and visually inspecting the resulting curves. The lowest points separating the peaks in the kernel density curve constructed with the selected optimal bandwidth were considered the separation points for instars. After separating the head capsule width data into subsets representing each instar, a further procedure similar to that in the first method was followed.

Approach 3 was a model-based clustering method based on a Gaussian mixture model for univariate data [25]. One assumption behind model-based clustering (hereafter called the clustering approach) is that the data are generated by a mixture of underlying probability distributions (normal, in our case), in which each component (instar) represents a different group or cluster. For univariate data, component distributions are characterized by their means and scalar variances. There are two model options: those for equal variances (E) and those for unequal variances (V). The ‘best’ model was estimated by fitting models with differing parameterizations and/or numbers of components to the data by maximum likelihood estimation and then applying the Bayesian information criterion (BIC) and the integrated complete-data likelihood (ICL) criterion [26,27]. The ‘mclust’ package, version 5.4.5, in the R environment [27] was used for data analysis. *D. pini* was assumed to have up to nine instars, so that all major peaks observed on the histogram could be considered a separate instar.

To evaluate whether the mean head capsule widths for each instar, determined by different approaches in our studies and those described in the literature, adhered to Brooks-Dyar’s rule, Brooks-Dyar’s ratios between successive instars were calculated using the following equation [28]:g*_i_* = µ*_i_*/µ_(*i* − 1)_,(1)
where g*_i_* is Brooks-Dyar’s ratio and µ and µ_(*i* − 1)_ are the mean widths of the head capsule for the *i*th and (*i* − 1)th instars, respectively.

Then, Crosby’s growth ratio (C*_i_*) was calculated as:C*_i_* = 100(g*_i_* − g_(*i* − 1)_)/g_(*i* − 1)_.(2)

A difference exceeding ±10% in successive growth ratios indicates significant deviation from Brooks-Dyar’s rule. When C*_i_* > 10, some instars might be missing; when C*_i_* < −10, there might be too many instars.

In addition, ln-transformed mean head capsule widths were linearly regressed on respective instars, and the coefficient of determination R^2^ was used to assess the model’s goodness of fit. The closer the coefficient value was to 1.0, the better the model fit the data, and the better the instar separation followed Brooks-Dyar’s rule [19]. All required calculations were performed in Microsoft Excel.

## 3. Results and Discussion

### 3.1. Visual Analysis of the Frequency Distribution and NLLS Estimation

The head capsule widths of *D. pini* larvae ranged from 0.96 to 6.48 mm. After various histograms with different numbers of intervals (bins) were constructed, the histogram with 45 categories was selected as the most visually appropriate. The interval width was equal to 0.1227 mm. The patterns of the head capsule width frequency for the larvae of different groups (lab, field–lab, and field) were rather similar (Figure 1); therefore, the data for these groups were combined.

Although the plot for the head capsule widths of all larval groups had 11 peaks (Figure 2), only 9 peaks and their respective ranges were identified as potentially corresponding to individual instars. Two peaks (between 3.29 and 3.41 mm and between 5.13 and 5.25 mm) were included in the fifth and eighth instars, respectively, because their heights were lower than those of the other peaks and the lowest points of the ranges were higher than those for the neighbouring peaks. Only the distributions of the first two instars were relatively narrow and non-overlapping. The distributions of the remaining instars overlapped to various extents, skewed to either side, or had multiple peaks that made instar separation difficult. The origin of the larvae could have a substantial effect on the frequency distribution of the head capsule widths. The first instar and a large portion of the second instar were measured only on larvae reared in the laboratory, whereas the other larvae originated both from laboratory rearing and field collections in different regions of Poland and in different years. The latter larvae were developed in various environmental conditions, fed on host plants of different qualities, and were of different sexes. All these factors have been shown to have a strong effect on the number of instars and the respective variation in head capsule widths in different insects [29,30,31].

The visually determined ranges of head capsule widths for different instars enabled the calculation of the initial values of the observed means and standard deviations (SDs) (Table 2). Then, these initial values were used for fitting functions to the frequency distribution of each instar (Figure 2). After the parameters were estimated by NLLS, the new combined function for all the instars was fitted (Figure 2), and the new values of the means, SDs, and ranges of head capsule widths for each instar based on the theoretical frequency distribution were calculated (Table 2). The estimated means did not differ from those observed, but the SDs and upper and lower limits did differ between the estimated and observed values. The smallest difference between the observed and estimated SDs was for the first and second instars that were the least overlapping (Figure 2).

### 3.2. Kernel Density and NLLS Estimation

The optimal bandwidth selection performed using the SRT and SJ methods resulted in band widths of 0.1841 and 0.0391 mm, respectively. The first method gave a significantly over-smoothed estimate, whereas the second method yielded a slightly under-smoothed density with too many noise-producing peaks, particularly after the third peak (Figure 3a,b, respectively). The tendencies of the SRT method to over-smooth and the SJ method to perform better with multimodal functions have also been observed in other studies [24,32]. However, the SJ method may perform poorly if the true density deviates from a normal distribution [33]. Visual assessment of the KDE function at different band widths ranging from 0.05 to 0.12 seemed to yield the best results, and the optimal band width was 0.08 mm because, at this level, the number of peaks seems to be reasonable and does not indicate over- or under-smoothing. This value was closer to the value estimated by the SJ method than that estimated by the SRT. Either 8 or 10 peaks could be separated visually; however, 8 peaks (Figure 3c) were selected, reflecting the most likely number of instars of *D. pini* larvae, while assuming that the peaks neighbouring the seventh peak could result from differences in larval development originating from different populations. For the frequency distribution generated by KDE with a band width of 0.08 mm, the interval width was 0.0119 mm.

The means and SDs (Table 2) calculated from the ranges of head capsule widths determined from the separation (lowest) points between each pair of consecutive instars (Figure 3c) were used to calculate the initial parameters for fitting functions to the frequency distribution of each instar. After NLLS estimation of the parameters, a new combined function was fitted to the data for all instars, and the new values of the means, SDs, and limits of the head capsule widths for each instar (Table 2) were calculated based on theoretical frequency distribution. The greatest changes between the observed and estimated values were observed in the last two instars.

### 3.3. Model-Based Clustering

Comparison of the BICs for the models with one to nine components (instars) revealed that the best models (i.e., those with the highest BIC values) were those with seven, eight, and nine components (Figure 4a).

The ICL values indicated that the best models should have a maximum of four components (Figure 4b); however, when the number of instars in *D. pini* larvae was assumed to be at least six, the best models should have eight or nine components. The lowest ICL values for the seven-component model indicate the very low ability of this model to classify observations into well-separated clusters [27]. Therefore, an eight-component model was selected for classification. This model suggested a rather wide range of head capsule widths for the first and eighth instars, and a relatively narrow range for the second instar (Figure 5). It is interesting that the number of suggested instars (i.e., eight) was higher than the number of peaks (i.e., six) on the density curve (Figure 5a). The mean for the sixth instar passed through a trough between the peaks, whereas the means for the seventh and eighth instars passed near each other through the same last peak.

The means and SDs of the head capsule widths for each instar were calculated from the model, and the ranges were determined based on the classification of the observed values (Table 2).

### 3.4. Adherence of the Mean Head Capsule Widths to Brooks-Dyar’s Rule

Two (KDE and clustering) of the three approaches applied in our study for *D. pini* larval instar separation resulted in eight distinguishable instars, and the other approach, i.e., the visual approach, produced nine instars (Table 2). These numbers are higher than those in the literature, i.e., seven instars (Table 1). The mean values of the head capsule width were most similar when estimated by the visual and KDE approaches. A notable difference can be observed only from the seventh instar, but it could be explained by the differences in the separated numbers of instars. The numbers of individuals included in the last instar in both approaches were smaller than those of the other instars, which means that this instar might be a supernumerary instar as a result of larval development in unfavorable conditions [30,31]. The means for the smallest first–third instars estimated by the visual and KDE approaches were similar to those presented by [13]. In all other cases, the mean values of the head capsule widths differed greatly.

Based on the mean values of head capsule widths estimated in our study and those values available in the literature, Brooks-Dyar’s ratios and Crosby’s growth ratios were calculated and are presented for comparison in Table 3.

Brooks-Dyar’s ratios were the most stable (1.22–1.36) for mean values estimated using the KDE approach. For the visual and clustering approaches, these ratios were stable until the sixth and fifth instars, respectively, and then dropped below 1.2 (Table 3). For the data reported in the literature, these ratios varied greatly: from 1.20 to 1.40 in [13], from 1.20 to 1.80 in [11], and from 1.13 to 1.50 in [14]. Our results do not support the general rule that the mean Brooks-Dyar’s ratio for hemimetabolous insects is close to 1.27 and that the ratio for holometabolous insects is approximately 1.52 [34].

Crosby’s ratios did not exceed 10% when calculated using the data in this study, regardless of the applied approach. In contrast, these ratios exceeded the 10% limit in data presented in the literature (Table 3), thus meeting a criterion of Crosby’s growth rule [19] and therefore indicating that suggested instar separations did not adhere to Brooks-Dyar’s rule. In most cases, the aberrant values were negative (from −10.42 to −25.93), which means that there might have been too many instars or that consecutive instars were not separated far enough from the previous instar. Crosby’s ratios were positive for the oldest instars only (16.83 and 32.83), indicating that at least one instar was missing.

It is interesting that all the approaches used in the current study seem to separate the instars well, as indicated by Crosby’s ratios, but produced different numbers of instars: the visual approach separated nine instars, and the two other approaches separated eight instars (Table 2), which may mean that when used alone, Crosby’s growth ratio is insufficient to properly evaluate whether the applied method correctly separates the instars. It is necessary to check additionally whether the means adhere to Brooks-Dyar’s rule by estimating and comparing the goodness of fit of a linear regression model of the ln-transformed mean head capsule widths on the instars. Among the three approaches used in this study, KDE gave the results with the best fit to a straight line (R^2^ = 0.997, Figure 6b). The visual approach separated too many instars at the end of larval development because the line connecting the estimated mean values turned slightly downward after the sixth instar, thus leading to a decrease in the R^2^ value to 0.980 (Figure 6a). Instar separation by the clustering approach was the least satisfactory (Figure 6c). Among the data in the literature, the regression line fit well to data presented by [13] (Figure 6d); however, in this case, Crosby’s growth ratios must be taken into account: three out of the five values exceeded 10% (Table 3), thus enabling an overall conclusion that the mean head capsule widths proposed by [13] did not adhere to Brooks-Dyar’s rule.

In summary, our results showed that more than one method should be used for instar separation in insects and that the one that enabled correct instar determination, verified using Brooks-Dyar’s rule, should be selected. Of the methods tested in this study, KDE was the best approach. In addition, both criteria, i.e., Crosby’s growth ratio < |10| and R^2^ > 0.99 for linear regression of ln-transformed head capsule widths on instars, must be met to conclude that instar separation based on the head capsule width adheres to Brooks-Dyar’s rule.

## 4. Conclusions

In this study, it was determined that *D. pini* larvae pass through eight instars during their growth, although the eighth instar might be a supernumerary instar as a result of larval development in unfavorable conditions. Of the three tested approaches (i.e., the visual approach followed by NLLS estimation, KDE followed by NLLS estimation and clustering), KDE produced the best results, which adhered to Brooks-Dyar’s rule by meeting both conditions: (1) Crosby’s ratio was less than 10%, and (2) the coefficient of determination R^2^ for a regression line fitted to the means plotted against the instars had the highest value, which was close to 1 (0.997). The new mean values and ranges of head capsule widths for each *D. pini* instar were determined, and those values and ranges estimated by KDE followed by NLLS can be used in further studies and for practical purposes. The number of instars and estimated mean values differed from those presented in three literature sources [11,13,14]. All the data available in the literature violated Brooks-Dyar’s rule because Crosby’s ratio exceeded 10% and the R^2^ values were low in two cases. The results showed that correct instar separation is possible only when more than one method is used for determining the number of instars and both conditions of Brooks-Dyar’s rule are met: (1) the regression line of ln-transformed mean head capsule widths on respective instars follows a straight line as closely as possible (i.e., R^2^ > 0.99) and (2) the deviation from a straight line measured by Crosby’s ratio should not exceed 10%.

## Figures and Tables

**Figure 1 insects-10-00384-f001:**
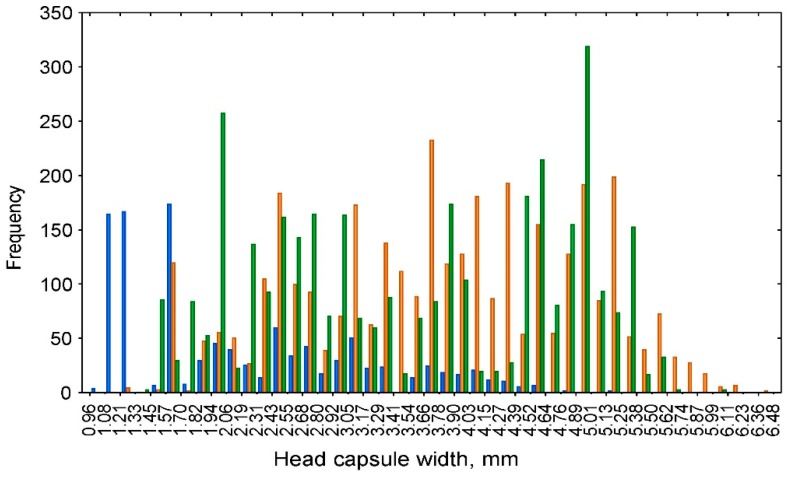
Frequency distribution of the head capsule widths for three groups of *Dendrolimus pini* larvae: lab (blue), field–lab (orange), and field (green).

**Figure 2 insects-10-00384-f002:**
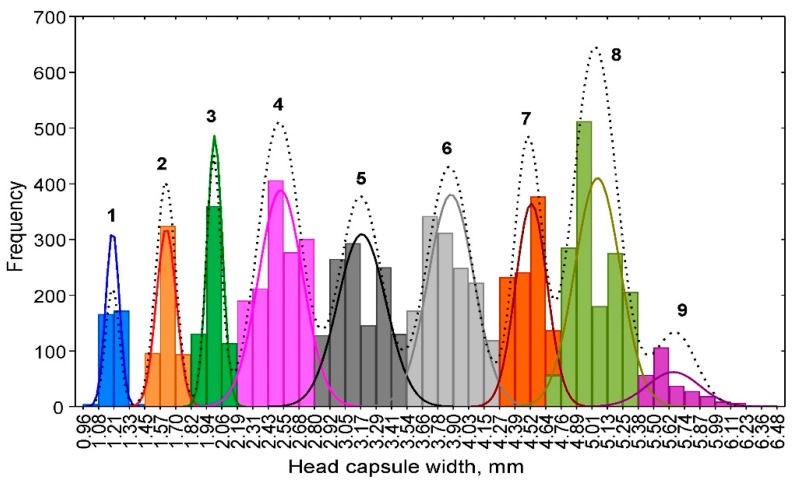
Frequency distribution of the observed *Dendrolimus pini* head capsule widths with fitted functions for each separate instar (the line color is the same as the column color) and the overall distribution based on non-linear least squares (NLLS) parameter estimates (black dotted line); the numbers indicate larval instars.

**Figure 3 insects-10-00384-f003:**
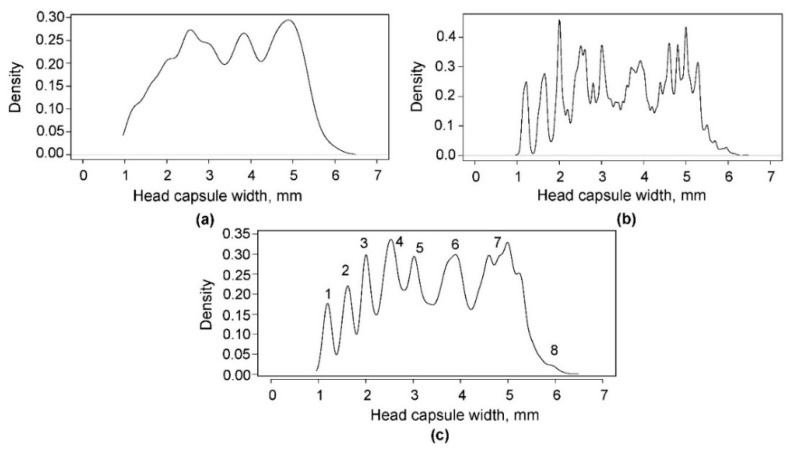
Kernel density of the *Dendrolimus pini* head capsule widths with the optimal band width estimated (**a**) using Silverman’s rule of thumb (band width = 0.1841), (**b**) using Sheather and Jones’ method (band width = 0.0391), and (**c**) visually (band width = 0.08); the numbers above the peaks indicate the most likely separation of larval instars.

**Figure 4 insects-10-00384-f004:**
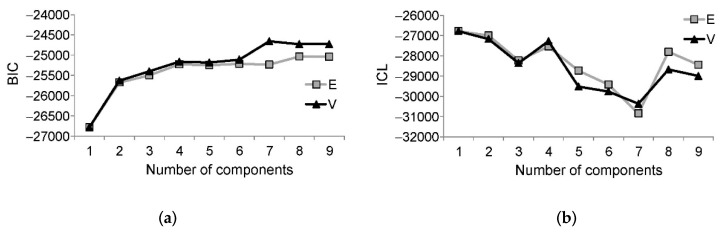
Bayesian information criterion (BICs) and integrated complete-data likelihood criterion (ICLs) for the Gaussian mixture models built for head capsule widths of *Dendrolimus pini* with various numbers of components (instars): (**a**) BIC and (**b**) ICL; E—Models for equal variances, V—Models for unequal variances.

**Figure 5 insects-10-00384-f005:**
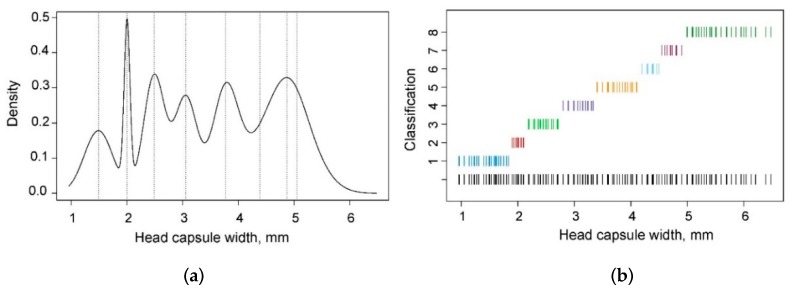
The results of clustering for the head capsule widths of *Dendrolimus pini* larvae based on the eight-component model (separating eight instars): (**a**) density (means are indicated by dotted lines) and (**b**) classification.

**Figure 6 insects-10-00384-f006:**
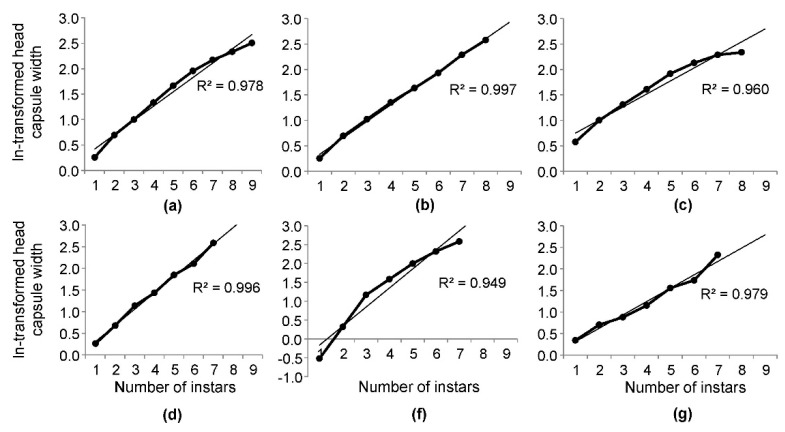
Observed changes of the ln-transformed head capsule widths of *Dendrolimus pini* larvae with their growth presented by increasing instar (bold solid line), fitted lines, and coefficients of determination indicating the goodness of fit based on (**a**) the visual approach and NLLS; (**b**) the kernel density estimation (KDE) approach and NLLS; (**c**) the clustering approach; (**d**) the data presented by Il’inskij and Tropin [13]; (**f**) the data presented by Śliwa [11]; and (**g**) the data presented by Pszczolkowski and Smagghe [14].

**Table 1 insects-10-00384-t001:** Mean head capsule widths of *Dendrolimus pini* larvae given in the literature.

Instar	Head Capsule Width (mm) According to		
	Il’inskij and Tropin [13]	Śliwa [11]	Pszczolkowski and Smagghe [14]
1	1.2	0.7	1.27
2	1.6	1–1.5	1.63
3	2.2	2–2.5	1.84
4	2.7	~3	2.22
5	3.6	~4	2.94
6	4.3	~5	3.33
7	6.0	6	5.01

**Table 2 insects-10-00384-t002:** Observed and estimated values for the means, standard deviations (SD), and limits of the head capsule widths (mm) of *Dendrolimus pini* larvae.

Instar	Observed			Estimated		
	Mean	SD	Limits	Mean	SD	Limits
Visual approach followed by NLLS estimation (nine instars)						
1	1.19	0.053	<1.45	1.19	0.053	<1.36
2	1.62	0.078	1.45–1.82	1.62	0.078	1.36–1.83
3	2.00	0.061	1.83–2.18	2.00	0.059	1.84–2.13
4	2.52	0.174	2.19–2.80	2.52	0.167	2.14–2.85
5	3.17	0.191	2.81–3.53	3.17	0.170	2.86–3.50
6	3.87	0.182	3.54–4.21	3.87	0.176	3.51–4.25
7	4.51	0.125	4.22–4.70	4.50	0.110	4.26–4.69
8	5.03	0.180	4.71–5.37	5.03	0.174	4.70–5.43
9	5.63	0.209	>5.37	5.67	0.167	>5.43
KDE approach followed by NLLS estimation (eight instars)						
1	1.19	0.048	<1.39	1.19	0.048	<1.36
2	1.62	0.080	1.39–1.80	1.62	0.080	1.36–1.80
3	2.03	0.091	1.81–2.22	2.03	0.089	1.81–2.23
4	2.56	0.150	2.23–2.81	2.55	0.143	2.24–2.83
5	3.11	0.148	2.82–3.41	3.11	0.135	2.84–3.39
6	3.82	0.216	3.42–4.22	3.81	0.196	3.40–4.18
7	4.88	0.345	4.23–5.70	4.89	0.339	4.19–5.80
8	5.94	0.143	>5.70	5.97	0.116	>5.80
Clustering approach (eight instars, limits are based on the classification of the observed values)						
1	-	-	-	1.49	0.255	0.96–1.84
2	-	-	-	2.00	0.052	1.90–2.10
3	-	-	-	2.48	0.183	2.19–2.71
4	-	-	-	3.05	0.212	2.80–3.33
5	-	-	-	3.78	0.226	3.40–4.11
6	-	-	-	4.37	0.321	4.20–4.50
7	-	-	-	4.88	0.312	4.55–5.08
8	-	-	-	5.05	0.417	5.10–6.48

**Table 3 insects-10-00384-t003:** Brooks-Dyar’s ratios (g) and Crosby’s growth ratios (C) for head capsule widths of *Dendrolimus pini* larvae at different instars separated using three approaches tested in our study compared to those presented by other studies (the bold values indicate C > |10|, indicating significant deviation from Brooks-Dyar’s rule).

Instar	Visual Approach and NLLS (Nine Instars)		KDE Approach and NLLS (Eight Instars)		Clustering Approach (Eight Instars)		Il’inskij and Tropin [13] (Seven Instars)		Śliwa [11] (Seven Instars)		Pszczolkowski and Smagghe [14] (Seven Instars)	
	g	C	g	C	g	C	g	C	g	C	g	C
1												
2	1.36		1.36		1.34		1.33		1.79		1.28	
3	1.24	−8.83	1.26	−7.44	1.24	−7.62	1.38	3.12	1.80	0.80	1.13	**−12.05**
4	1.26	2.04	1.26	−0.10	1.23	−0.82	1.23	**−10.74**	1.33	**−25.93**	1.21	6.88
5	1.26	−0.31	1.22	−3.02	1.24	0.77	1.33	8.64	1.33	0.00	1.32	9.76
6	1.22	−2.67	1.23	0.80	1.16	−6.72	1.20	**−10.42**	1.25	−6.25	1.13	**−14.47**
7	1.16	−5.04	1.28	4.46	1.12	−3.41	1.40	**16.82**	1.20	−4.00	1.50	**32.83**
8	1.12	−3.70	1.22	−4.70	1.03	−7.33	-	-	-	-	-	-
9	1.13	0.69	-	-	-	-	-	-	-	-	-	-

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
