# Peer review of "A Comparison of Three Approaches for Larval Instar Separation in Insects—A Case Study of Dendrolimus pini"

_insects, 2019, doi:10.3390/insects10110384_

Round 1
Reviewer 1 Report
The manuscript (Determining the number of Dendrolimus pini larval instars – a comparison of three approaches (insects-610796)) deals with the determination approaches for number and dimensions of instar of important insect pest for conditions of central Europe.
The authors have done a relatively large set of experiments and data analysis and obtained corresponding results, especially in the determination of the best approach for instar determination by measuring and statistical analyses of caterpillar head capsule width.
There are (please see below) some suggestions for better intelligibility of text (especially in part Introduction and Conclusions, but not only).
Questions, comments:
The title could better describe the content of the whole manuscript and correspond to the goals. The article does not only about the determination of the number of instars.
The abstract is written very well, but setting individual study goals using numbering ("1) and 2)") is inappropriate.
Line 26. It is reported that D. pini is one of the major defoliators of pine. I would recommend extending the sentence by the fact that only caterpillars of this kind are phytophagous.
Lines 27, 28 and 29. Distribution of D. pini is reported here. It would be useful to add citations to the publications which were used by the author when writing this part of the manuscript.
Line 35. The author describes "this species". Please explain which species.
Line 35-43. Here are details of D. pini bionomy. It would be useful to add citations to the publications which were used by the author when writing this part of the manuscript.
Line 50. In Table 1 (also line 270) I would prefer to quote in Latin characters. E.g. Cyrillic should not also be used in References (lines 307-390). The possibility how presented information about e.g. Russian language in reference is following "(In Russian or etc.)".
Line 58. There is a clearly missing part in the sentence, probably the word "and". It is part of the sentence „are observed „and“ when the“.
Line 59-61. It would be useful to add citations to the publications which were used by the author when writing this part of the manuscript.
The part “Materials and Methods”. Here it would be appropriate to add parameters, i.e. temperature, humidity and light conditions of the breeding in which the caterpillars were kept.
Line 77. Please explain which instars the author has described as younger and older and whether the selection matches other parts of the text (for example, lines 150, 226, etc.)
Line 147. It would be useful to add citations to the publications which were used by the author when writing this part of the manuscript.
Line 154. There is redundant space in the parentheses at the end of the sentence (“[27- 29]”).
Figures: All charts have reduced quality in the final version of the manuscript. Therefore, please add a better (quality) and the same format of all graphs (font size in each graph, etc.).
Line 218. It would be appropriate to move the headline to the next page, this form is not appropriate.
Line 126. The word "supernumerary" should be replaced by a synonym.
Figure 5. I would adjust this graph from 2x2x2 layout to 3x3. This will be appropriate in terms of a clear distinction between the data collected and the cited research data.
Line 393-297. I would adjust this text for clarity. The point is that the author clearly determined the optimal procedure for the determination of instars and then concludes that: „The results showed that correct instar separation is possible only when more than one method is used for determining the number of…“.
Reference. This part needs to be supplemented by a very detailed revision - above all, it is important to unify the style of all references (dashes, spaces, etc.).
I do not recommend the manuscript publication in Insects in its present form, but it will be appropriate after minor revision.
Reviewer 2 Report
Although the author has presented her claims clearly, I feel the data presentation should be improved. It is because I am not sure whether the lab colony and wild population are comparable. For instance, it is known that temperature strongly affects body size in insects, meaning head capsule width must be also affected. Therefore, I strongly recommend to show lab colony and wild population results separately as well as two together. In addition, the author must clearly describe how the lab colony was maintained.
Minor point. This is a single author paper. Why do you say "we" instead of saying "I"? If the author feels "we" is appropriate, you must include others as authors.
Reviewer 3 Report
The author studied the “Determining the number of Dendrolimus pini larval instars – a comparison of three approaches”. While I have no objection against publishing the data, I have some issues that need addressing. I am concerned about the poorly elements in the methods; the author must be provided details for results interpretation. The experimental set up of this study appears to be well-designed and the data collected carefully. However, information about some methods and results obtained in the experiments need to be clarified. My specific comments are listed in the "Comments and Suggestions for Authors
". Based on the comments above reported, my opinion is that this manuscript may be suitable for printing on this journal after corrections.
Please see my specific comments below:
L.12: Delete “Generally”
L.13-16: Summarize the objectives into a single objective
L.22-23: Keywords should be in alphabetic order.
L.27:...important defoliating pest of Pinus sylvestris L. in Europe and Asia.
L.32: Change “abundance” by “populations”
L.39: ...they pupate in June. Any references?
L.40-42: Sentence contains repetitive words, rewrite.
L.50: In all tables, provide the full scientific name. Also, The reference "Ильинский и Тропин" should be in English.
L.72: Sentence starting “Three types of…”
L.76: What were the laboratory conditions? How were the insects fed? Please, explain.
Please clarify why you measured individuals from larva stages
L.80: How often was the head capsule measured? (hours? days?)
L.141: Delete “approximately”
L.142: Delete “approximately”
L.272-274: Delete this sentence.
L.290-297: Please, define and conclude which is the best method according to the results of the study.
Round 2
Reviewer 3 Report
The manuscript “A comparison of three approaches for larval instar separation in insects – the case study of Dendrolimus pini” has been improved and all my questions were taken into account. I recommend the publication in “Insects”.